# Increased Co-Occurrence of Pathogenic Variants in Hereditary Breast and Ovarian Cancer and Lynch Syndromes: A Consequence of Multigene Panel Genetic Testing?

**DOI:** 10.3390/ijms231911499

**Published:** 2022-09-29

**Authors:** Mar Infante, Mónica Arranz-Ledo, Enrique Lastra, Luis Enrique Abella, Raquel Ferreira, Marta Orozco, Lara Hernández, Noemí Martínez, Mercedes Durán

**Affiliations:** 1Cancer Genetics Group, Excellence Unit of the Institute of Biology and Molecular Genetics, University of Valladolid-Spanish National Research Council (IBGM, UVa-CSIC), C/ Sanz y Forés 3, 47003 Valladolid, Spain; 2Unit of Genetic Counseling in Cancer, Burgos University Hospital, 09006 Burgos, Spain; 3Unit of Genetic Counseling in Cancer, Rio Hortega University Hospital, 47012 Valladolid, Spain

**Keywords:** Hereditary Breast and Ovarian Cancer Syndrome (HBOC), Lynch Syndrome (LS), multi-gene panel testing, double heterozygotes, genetic counseling

## Abstract

The probability of carrying two pathogenic variants (PVs) in dominant cancer-predisposing genes for hereditary breast and ovarian cancer and lynch syndromes in the same patient is uncommon, except in populations where founder effects exist. Two breast cancer women that are double heterozygotes (DH) for both *BRCA1/BRCA2*, one ovarian cancer case DH for *BRCA1/RAD51C,* and another breast and colorectal cancer who is DH for *BRCA2/PMS2* were identified in our cohort. Ages at diagnosis and severity of disease in *BRCA1/BRCA2* DH resembled *BRCA1* single-carrier features. Similarly, the co-existence of the *BRCA2* and *PMS2* mutations prompted the development of breast and colorectal cancer in the same patient. The first *BRCA1/BRCA2* DH was identified by HA-based and Sanger sequencing (1 of 623 families with BRCA PVs). However, this ratio has increased up to 2.9% (1 DH carrier vs. 103 single PV carriers) since using a custom 35-cancer gene on-demand panel. The type of cancer developed in each DH patient was consistent with the independently inherited condition, and the clinical outcome was no worse than in patients with single *BRCA1* mutations. Therefore, the clinical impact, especially in patients with two hereditary syndromes, lies in genetic counseling tailor-made for each family based on the clinical guidelines for each syndrome. The number of DH is expected to be increased in the future as a result of next generation sequencing routines.

## 1. Introduction

Among 5–10% of all breast and ovarian cancers (BOCs) are considered to be hereditary, and about 30% of them are due to pathogenic mutations in the two major susceptibility genes, *BRCA1* and *BRCA2* [1]. These patients have an increased risk of developing cancer at an early age, as well as multiple primary cancers. Lynch syndrome (LS) is responsible for about 1–3% of all colorectal cancer (CRC) cases. Germline variants in any of the DNA mismatch repair (MMR) genes, as well as germline deletions of the EPCAM gene, are the main cause of this syndrome [2].

The frequency and incidence of mutations in these genes varies according to geographical area or the ethnicity, in addition to other non-genetic factors. The probability of carrying both *BRCA1* and *BRCA2* mutations in a single patient is an uncommon phenomenon, except in certain subpopulations where founder effects exist. It has been reported that the percentage of double heterozygotes (DH) varies between 0.2 and 0.8% in different ethnic groups, whereas it reaches 1.8% in Ashkenazi Jewish people [3]. Around thirty studies have previously identified BOCs families that harbor mutations in both *BRCA1* and *BRCA2* [4].

Traditionally, probands who meet the risk criteria for a given inherited syndrome undergo germline testing, targeting the specific genes associated with each syndrome [5]. Whenever a germline pathogenic variant (PV) is found, targeted therapies in patients or inclusion in cancer surveillance programs for their relatives are enabled. Since next generation sequencing (NGS) technologies arose to analyze multiple hereditary cancer genes at the same time, with less time consumption, and at a lower cost, the use of multigene panel testing has grown over the last 10 years, which has led to an increased identification of individuals that carry two or more inherited cancer predisposition alleles [6]. As the number of genes included in a panel expands, this phenomenon is expected to increase. In fact, Whitworth and colleagues named it Multilocus Inherited Neoplasia Alleles Syndrome (MINAS) [7]. Until recently, a low frequency of MINAS cases had been reported, mainly due to previous diagnostic procedures in which the phenotype was attributed to the first identified PV; therefore, MINAS cases are likely to be the cause of distinct and overlapping phenotypes [6,7].

Dealing with clinical heterogeneity is a great challenge for genetic counselling in hereditary cancer syndromes. The variability in the phenotype could be explained by several factors, such as incomplete penetrance, allelic and genetic heterogeneity, the presence of genetic modifiers, and even environmental factors.

Nevertheless, it is unclear whether a more severe phenotype, such as earlier-onset or more aggressive tumors or even other unrelated cancer types, would be observed in families with overlapping criteria for several genetic cancer syndromes. Hence, the aim of this study was to add insight into the prevalence of the co-occurrence of germline pathogenic variants (PVs) in breast and/or ovarian cancer syndrome (HBOC) and LS syndromes and to characterize the phenotypes of our probands in order to achieve a more accurate assessment of the risk of cancer in families with two PVs. Herein, we report four families with germline PVs, conferring high to moderate risk in two dominant cancer-predisposing genes (CPGs) for HBOC and LS syndromes.

## 2. Patients and Methods

Families fulfilling the selection criteria defined in the SEOM (Spanish Society of Medical Oncology): clinical guidelines for HBOC [8] or LS [9] were referred for BRCA or MMR genetic testing from our Cancer Genetic Counseling Units (CGUs) of the Hereditary Cancer Program of the Regional Government of Castilla and León (Spain). The eligible individual for genetic testing was, if possible, the youngest breast cancer (BC), ovarian cancer (OC) or colorectal cancer patient, but mutation screening was occasionally offered to a healthy person if there were no cases of BC, OC, or CRC available for testing. EDTA-anticoagulated peripheral blood samples, pedigrees with family history, and clinical features of cancers, as well as the informed consent of index cases or relatives, were sent to the IBGM laboratory for BRCA or MMR genetic testing. All participants signed the informed consent for genetic diagnosis and agreed to participate in any future research project related to hereditary cancer. This study was performed in line with the principles of the Declaration of Helsinki. Approval was granted by the Ethics Committee of Clinical Research of the health areas of Burgos and Soria.

## 3. Mutation Analysis

DNA and RNA were extracted from peripheral blood as described elsewhere [5,10]. The mutational analysis of codifying regions of predisposing genes was performed with the screening method used in our diagnostic routine in each period. Until 2013, we used CSGE, HA-CAE [11] or HRMA [5] followed by the Sanger sequencing of those fragments with an altered mobility pattern (herein, HA-based methods). From 2014 onwards, we have used NGS equipment for genetic testing in HBOC families: firstly, GS Junior with Multiplicom BRCA MASTR Dx with 454 MID Dx kit and, later, Illumina MiSeq with Multiplicom BRCA MASTR Dx with drMID Dx. In 2018, we began using the Ion Torrent™ Technology using the Oncomine™ BRCA Assay and a 35-gene Ion Ampliseq on-demand panel to explore HBOC and LS on the Ion S5™ System (the list of 35 genes was described by Velazquez et al. [10]). The Library and template preparation was automatically carried out with the Ion Chef™ System and loaded in an Ion 520 Chip, as described in a previous report [10]. The Ion Reporter software (Version 5.10) was used for filtering and variant annotation. Although a minimum coverage of 30× was considered, the mean percent target coverage at 50× was 88.6% and the coverage uniformity was higher than 90% in all tested samples [10].

All pathogenic variants (PVs), or likely pathogenic variants (LPVs) (hereinafter PVs), were confirmed on a second DNA sample by Sanger sequencing with the BigDye Terminator Sequencing Kit v3.1 (Applied Biosystems, Foster City, CA, USA) on an ABI 3130XL DNA Sequencer. Whenever possible, the segregation of the PVs was also performed in the probands’ relatives by Sanger sequencing.

All NGS Technologies used have the capacity to identify large rearrangements (LGRs), except the Ion AmpliSeq on-demand panel. To carry out the detection of LGRs in the on-demand panel-tested samples, we used the Multiplex Ligation Dependent Probe Amplification (MLPA) (MRC Holland) according to the manufacturer’s instructions. MLPA probe mixes: *BRCA1* (P002), *BRCA2* (P045), *MLH1*/*MSH2* (P003), *PMS2* (P008), *MSH6*/*MUTYH* (P072), and MMR methylation (ME011).

Mutations were numbered according to HGVS nomenclature guidelines for cDNA sequences.

## 4. Results

Firstly, a total of 2623 families meeting the HBOC and LS criteria were analyzed with HA-based methods. We identified 794 PVs in BRCA (623) or MMR genes (171), finding a family that carried two mutations in each *BRCA1* and *BRCA2* genes (0.16%). Secondly, we performed NGS in a total of 2121 families fulfilling the criteria for HBOC (1729 families) or LS (392 families), 414 of whom were run with the on-demand 35-gene panel (163 HBOC and 251 LS). We identified PVs in a total of 177 families analyzed with NGS, 116 of whom were identified by the S5 Ion Torrent. Focusing on those samples discovered by the Oncomine Panel, 53 families were carriers of PVs in 13 non-BRCA/MMR genes. Among them, we observed three more women with two mutations in *BRCA1*/*BRCA2, BRCA2*/*PMS2* and *BRCA1*/*RAD51C*. All these mutations had previously been submitted as PVs or LPVs in public variant databases such as ClinVar, LOVD, BRCA Exchange, or CanVarUK.

### 4.1. Family 678

The proband (II.2) (Figure 1A and Table 1) was a 72-year-old woman diagnosed with infiltrating ductal carcinoma breast cancer (IDC-BC) at 48 years of age. Immunohistological evaluation showed the negative expression of hormonal estrogen (ER) or progesterone (PR) receptors. She underwent prophylactic mastectomy and risk-reducing salpingo-oophorectomy (RRSO) accompanied by hysterectomy. Now, she is cancer-free. Several phenotypes were observed in this family (Figure 1A and Table 1). Her daughter (III.2) was a 34-year-old-woman affected with stage IIIc bilateral papillary serous ovarian carcinoma (HGSC). She embarked on infertility treatment six months before diagnosis and passed away from the disease at age 36. Other tumors referred by this family were three gastric cancer cases and one CRC. The mutational analysis showed that the proband carried two PVs: c.34C>T in the *BRCA1* gene that causes a premature stop codon (p.Gln13Ter) and a second frameshift mutation in the *BRCA2* gene c.1587delTinsCA (p.Glu532ArgfsTer3). Intriguingly, both mutations were identified in the index case, but only c.34C>T was detected in her daughter, diagnosed with OC (Figure 1). The husband of the proband (II.1) did not carry any of the PVs. Several relatives were also tested in this family. A 52-year-old niece (III.3) who developed hormone receptor-positive, HER2-negative IDC-BC at age 40 only harbored the *BRCA2* mutation. Consequently, her father (II.4), who suffered from gastric cancer at the age of 54, is an obligate carrier. This woman is disease-free to date, probably due to prophylactic surgery. The son of the index case (III.1), also DH, is currently healthy under prostate surveillance.

### 4.2. Family 776

The index case was a woman who developed BC at 40 years old (Figure 2A and Table 1); the tumor type was IDC hormone-sensitive (ER-positive/PR-positive) and human epidermal growth factor receptor type 2 (HER2)-negative. Molecular analyses revealed that she carried a PV c.5146_5149delTATG (p.Tyr1716LysfsTer8) in the *BRCA2* gene, so the analysis was extended to other relatives. Her parents and siblings were tested for the mutation, which confirmed that it was inherited from her father (II.7), who developed metastatic prostate cancer (PrC). Although three of her siblings were carriers of the *BRCA2* mutation, only one sister (III.4) has opted for prophylactic bilateral adnexectomy and mammographic surveillance. Recently, at the age of 54, the proband developed CRC, and, since her tumor showed a loss of expression in *PMS2*, we re-analyzed the DNA sample with our on-demand NGS panel. As a result, we discovered that she had the mutation c.903G>T (p.Lys301Asn) in the *PMS2* gene. Subsequently, we investigated whether the rest of the available family members were also carriers of the mutation, which is classified as an LPV in ClinVar. In this case, the mutation was maternally inherited, and all siblings harbored the LPV in *PMS2*. To date, all of them are disease-free. Her mother is under colonoscopy surveillance, where polyps are periodically resected.

### 4.3. Family 3699

The proband was a 67-year-old woman who was referred for genetic testing because she was diagnosed with metastatic IDC-BC in her mid-forties; immunochemistry showed negative ER/PR (Figure 1B and Table 1). Five years after primary systemic treatment for BC, she underwent hysterectomy and bilateral adnexectomy surgery. Both parents have had cancer, her mother specifically suffered from BC and her father suffered from lung cancer; unfortunately, both are dead so we could not see who she inherited the mutations from. Five of her eight siblings had been diagnosed for different types of cancer (OC, CRC, glioblastoma, or head and neck). Analysis with the Oncomine BRCA revealed that she carried two PVs in both *BRCA1* and *BRCA2*. The *BRCA1* mutation was a large genomic rearrangement (LGR) involving a complete deletion of *BRCA1*, an outcome that was also confirmed by MLPA analysis. The frameshift *BRCA2*-c.5796_5797delTA (p.His1932GlnfsTer12) variant was also present in the proband. Nevertheless, none of the affected relatives have undergone genetic testing so far.

### 4.4. Family C1423

The proband was an 86-year-old woman diagnosed with HGSC at the age of 80 who attended the Genetic Counselling Unit for suspicion of LS in her family (Figure 2B and Table 1). The patient mentioned numerous cases of cancer in her family: her father died of pancreatic cancer (PaC), three of her siblings have a personal history of CRC, and another has a history of PrC. Genetic testing with an NGS panel revealed that she had two PVs: *BRCA1*-c.4165_4166delAG (p.Ser1389Ter) and *RAD51C*-c.709C>T (p.Arg237Ter). Furthermore, two variants of uncertain significance (VUS) were identified in this proband: c.968A>G p.Lys323Arg) in the *BLM* gene and c.976G>A p.(Val326Met) in the *MLH1* gene. None of her relatives have so far been tested.

## 5. Discussion

The global frequency of PVs carriers in the BRCA1/2 genes is estimated to be 0.51%, and that in any of the MMR genes is around 0.36% [12]. Therefore, the probability of finding double PV carriers in the same patient is a rare phenomenon, except in certain ethnicities where there are founder effects, such as Ashkenazi Jews [12]. The number of publications reporting DH has noticeably increased, especially since 2016, when NGS was adopted as the technique to detect mutations in cancer-predisposing genes. Indeed, only the first DH family was identified when standard clinical practice was to scan *BRCA1* and *BRCA2* with HA-based methods until a PV was detected. Furthermore, the larger the number of genes included in the panels, the greater the amount of DH cases to appear; this was the case with our cohort, in which three more DH were identified since we adopted NGS.

Recently, McGuigan et al. [6] reviewed the co-occurrence of PVs, enclosed within a term coined as MINAS in 2016. In the aforementioned study, 385 individuals with more than two PVs in actionable genes, implying hereditary syndromes, were considered to establish phenotypic associations. Up to 78.5% of the cases contained at least one PV in *BRCA1* and/or *BRCA2*. Here, we report on four families with two germline PV/LPVs in genes that greatly increase the risk of HBOC or LS. Two families were double heterozygotes in both *BRCA1/BRCA2*, one family is *BRCA2/PMS2* heterozygote, and the last family was *BRCA1/RAD51C* heterozygote. BRCA genes and *RAD51C* confer high penetrance for HBOC, whereas *PMS2* is a predisposition gene for colorectal cancer. DH in *BRCA1* and *BRCA2* are present in 0.9% of breast (62/6545) and 0.9% of ovarian (17/1864) cancer cases of the international CIMBA dataset [13]; these figures rise to 1.8% (2/106) if populations with founder mutations are considered [3]. It was not until we began using NGS and implemented the on-demand 35-gene panel as a routine test for HBOC and LS that an additional 3 DH were detected versus 103 cancer patients (BC, OC or CRC) with a single PV in any of the genes included in the panel, which makes the percentage of DH in our population unusually high (2.9%). This elevated frequency could partially be attributed to the presence of founder mutations [14] or other recurrent PVs. In fact, the BRCA mutations in families C1423 and 678 were quite frequent in our series (Table 1); another possible cause is that families present more and more overlapping phenotypes between HBOC and LS instead of an isolated condition. Consequently, as more exome or genome sequencing outcomes become available, an increasing number of individuals with more than one PV will be reported, not only for those with *BRCA1*/*BRCA2* but also for other CSG combinations.

### 5.1. BRCA DH

The first DH identified in our series was a BC patient carrying *BRCA1*-c.34C>T and *BRCA2*-c.1587delTinsCA (Figure 1A). Interestingly, her daughter (III.2) with OC only inherited the *BRCA1* mutation, while the BC (III.3), a gastric cancer (II.4), and CRC (II.8) are the phenotypes only associated with the *BRCA2* mutation. *BRCA1*-c.34C>T is a mutation reported worldwide in individuals affected with BC and/or OC, whereas the *BRCA2*-c.1587delTinsCA mutation by itself was found in four more BC or OC families of our series. This *BRCA2* mutation has only been found in the Spanish and Hispanic population [15], so it could be a founder mutation.

Secondly, a complete deletion of *BRCA1* and the frameshift *BRCA2*-c.5796_5797delTA was found in the proband of the 3699 family. She only developed BC, most likely because she underwent risk-reducing surgery. Segregation in her relatives with CRC and PrC could not be ascertained. To date, around 25 cases of *BRCA1* deletion have been reported, [16], and most of them are of Spanish and Hispanic origin. *BRCA2*-c.5796_5797delTA has mostly been reported in Italian and Greek families with cases of BC, OC, or PaC. Additionally, this variant was found to co-occur with the PV c.5266dupC in *BRCA1* in two studies in women with BC and BOCs [17,18]. The coexistence of LGR in *BRCA1* and *BRCA2* PV is an extremely rare event; to our knowledge, only a single case of this has been described to date [13].

### 5.2. BRCA and Non-BRCA DH

Family 776 is a paradigm of the complexity of genetic risk management in cancer. LS and HBOC syndromes coexist in the same individual. In a first instance, genetic testing pointed out the mutation c.5146_5149delTATG in *BRCA2* as the cause of the BC in the proband diagnosed at 40 years of age. The mutation was inherited from her father, who had metastatic PrC. When she was diagnosed with CRC 14 years later, we discovered the *PMS2*-c.903G>T PV, which was maternally inherited. Her 82-year-old mother remains healthy despite the removal of some colorectal adenomas. *BRCA2*-c.5146_5149delTATG is a founder mutation in our region, accounting for 13.6% of all our *BRCA2*-positive families [14]. In our cohort, we observed this mutation in patients with BC, including bilateral and male, OC, PrC, CRC, and PaC [14,19]. Recently, this mutation was also found in DH with another *BRCA1* PV in an Argentinian family [20]. *PMS2* c.903G>T is a missense mutation located at the last nucleotide of exon 8 that causes aberrant splicing in minigenes [21] and leads to out-of-frame skipping. It has already been described in LS-associated cancer families and in constitutional mismatch repair deficiency syndrome. As far as we know, only one more case with two PVs in *PMS2* and *BRCA2* has been described in a Chinese woman who suffered from TNBC at 64 [6]. Mutations in *PMS2* are associated with the lowest risk (22%) and a higher average age of diagnosis (61–66 years) for any LS-related cancers [22,23]. Some researchers, however, have noticed greater percentages of *PMS2* mutations in the general population and in patients who meet screening criteria for HBOC but not LS [24,25]. Current knowledge suggests that mutations in *PMS2* marginally increase the risk of suffering BC (8–13% risk by age 70) compared with the general population. Unfortunately, we were unable to work out whether the *PMS2* mutation triggered the two cases of BC on the maternal branch of the family [24]. As a result, in this family in which two hereditary conditions (HBOC and LS) concur, the presence of the two PVs has prompted surveillance and prevention measures in healthy carriers according to the clinical guidelines for both syndromes.

The last proband carried two PVs in *BRCA1* and *RAD51C* and two VUS in *BLM* and *MLH1*. *BRCA1*-c.4165_4166delAG was found to be recurrent among our cohort (six families overall) and has frequently been identified worldwide in BOCs families, as well as in a gastric cancer case in our series. Likewise, the phenotypes associated with *RAD51C*-c.709C>T include BC, OC and metastatic PrC cases [26,27,28]. Furthermore, a triple heterozygosity case has been described in an HGSC woman who harbored two recurrent PVs in the Finnish population in the *RAD51C* and *ATM* genes together with a PV in the *BRCA1* gene [6,29]. Otherwise, two DH cases with different mutations in *BRCA2* and *RAD51C* have been described in Pakistani and Greek women who developed BC at 38 years and bilateral BC at 46 and 56, respectively [30,31].

In contrast to the previous family, *BRCA1* and *RAD51C* are associated with HBOC. It is well-established that PVs in *BRCA1* confer an increased risk of OC at an early age. Nevertheless, the age at diagnosis in this patient is more consistent with the mean age at diagnosis of those women with PVs in other susceptibility genes, such as *BRIP1*, *RAD51C*, or *RAD51D*—even with those defined for sporadic cases (63 years or older) [32]. Intriguingly, she was diagnosed at 81 years of age, very much older than the median ages for *BRCA1*-positive or *RAD51C*-positive women (53 and 61 years old, respectively) [32]. This patient is also a carrier of two VUS in *BLM* and *MLH1*, so we cannot rule out whether these VUS variants or other external factors might play roles as modifiers of the phenotype.

### 5.3. Impact of DH on Phenotype

Some attempts have been made to unravel the effect of concurrent mutations in *BRCA1* and *BRCA2* on the phenotype. On the one hand, it has been discussed whether a more severe phenotype or earlier onset of cancer could be produced in DH than carriers of PV in only *BRCA1* or *BRCA2* [6,13]. Both studies concluded that there were no significant differences in the mean age of BC diagnosis in DH compared to single carriers. A review of MINAS cases [6] investigated the synergistic or additive effects on the phenotype in 385 individuals reported in the literature, 287 of whom harbor P/LP on *BRCA1* and/or *BRCA2*. To support the synergy, the researchers speculated whether a more adverse phenotype would be observed in cases of PV carriers in cancer susceptibility genes that map to the same chromosomal region, since the loss of heterozygosity in the tumor would lead to a homozygous null. In such a case, our *BRCA1/RAD51C* patient would have developed cancer at a younger age than 81 years. In addition, our DH patients did not show an earlier age of cancer onset or an aggressive phenotype, rather the opposite: our DH patients were diagnosed with BC at 45 and 48 years old. Conversely, they do not show a more severe phenotype, such as an increased risk of developing another primary BC and/or OC, as has been suggested [3]. This is perhaps because they both underwent prophylactic surgery, which makes it unlikely that they will develop other tumors. On the other hand, and regarding the status of BC hormone receptors, the authors of the CIMBA study [13] already noted differences between DH as to *BRCA1* or *BRCA2* cases alone, setting the hormone receptor-associated phenotypes in DH midway between *BRCA1* or *BRCA2* only BC. The BC probands in families 678 and 3699 were found to be hormone-receptor-negative, but we cannot rule out whether both DH cases will be triple-negative because they were diagnosed in the 1990s when this biomarker was unusual in clinical practice. Altogether, it looks like phenotypes in DH more resemble the *BRCA1*-only BC [6,13], supporting an independent effect. Further evidence of additive consequences can also be seen in the *BRCA2/PMS2* carrier, who suffered from both BC and CRC.

The authors of some publications have even questioned whether BC should be included in the spectrum of LS tumors since *PMS2* and *MLH6* carriers are more likely to meet only BRCA1/2 criteria and not LS [24]. Thus, the significance of reporting PVs’ co-occurrence should be considered as an explanation for the phenotype displayed in some cases such as our *BRCA2*/*PMS2* patient. Overall, our results suggest that, in most cases, the effects are likely to be additive, which is consistent with the results of previous studies [6,12]. Despite the small number of patients reported here, these findings shed light on the phenotypic implications for DH carriers in clinically actionable cancer genes, and, as suggested by others [6], disclosing the genetic and clinical data of DH patients is key to establish relationships of double mutations in cancer-associated genes with worse clinic or an early onset of cancer. Nevertheless, our results should be validated in larger cohorts for more precise results.

Furthermore, the presence of several PVs in a patient involves a change when counselling first-degree relatives, mainly because they face a potential higher risk than if they harbored a single PV; otherwise, preventive management would not be the same if only one of the family mutations was considered.

## 6. Conclusions

We outline the phenotype displayed in women harboring two deleterious mutations in high-risk cancer-associated genes where the clinical outcome in DH carriers is no worse than in patients with single BRCA mutations. The type of cancer in each patient was found to be consistent with the inherited condition, reinforcing the hypothesis that two PVs play an independent role in the final phenotype. For this reason, the implications for follow-up and prophylactic measures recommended by clinical guidelines will not differ in cases of MINAS associated with HBOC syndrome, while tailored surveillance should be offered to those DH in the BRCA1/2 and MMR genes.

The value of genetic panels lies in their ability to engender the greater identification of patients carrying more than one germline PV, which is expected to be further increased with genome-wide studies. In addition, the characterization of DH individuals is needed to offer the most appropriate and personalized treatments and surveillance options for each cancer patient in the near future.

## Figures and Tables

**Figure 1 ijms-23-11499-f001:**
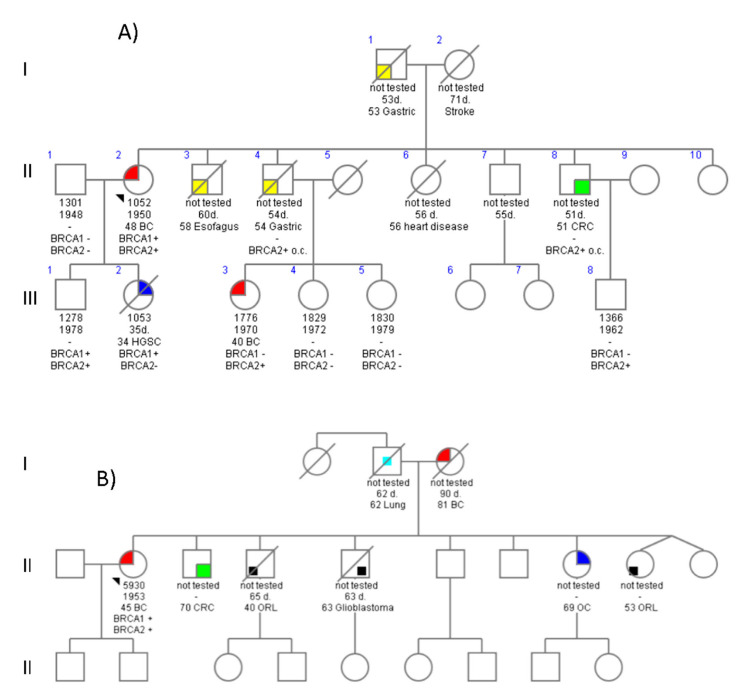
Pedigrees of double heterozygote *BRCA1/BRCA2* families. (**A**) Family with c.34C>T (p.Gln13*) in *BRCA1* and c.1587delTinsCA (p.Glu532Argfs*3) in *BRCA2.*; (**B**) Family with c.-232_*1383{0} (p.Met1ValfsX13) in *BRCA1* and c.5796_5797delTA (p.His1932Glnfs*12) in *BRCA2.*; Index cases are indicated by an arrow. Confirmed mutation carriers are indicated by a “+” sign, and non-carriers are indicated by a “-” sign; o.c., obligate carriers. Age at diagnosis and cancer type is specified as follows: BC, breast cancer; OC, ovarian cancer; HGSC, high-grade serous-papillary ovarian carcinoma; CRC, colorectal cancer; ORL, oral cancer. The colors stand for each type of cancer as follows: red stands for BC, blue stands for OC, green stands for CRC, yellow stands for Gastric Cancer, black stands for another type of cancer.

**Figure 2 ijms-23-11499-f002:**
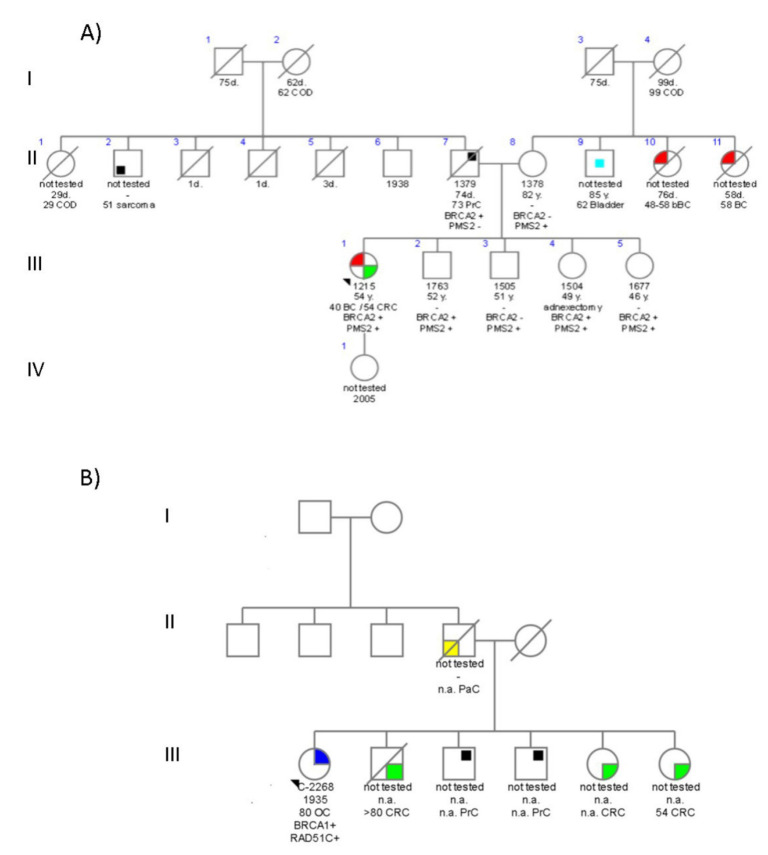
Pedigrees of double heterozygote *BRCA1/RAD51C* and *BRCA2/PMS2* families. (**A**) Family with c.5146_5149delTATG (p.Tyr1716Lysfs*8) in *BRCA2* and c.903G>T (p.Lys301Asn) in *PMS2*; (**B**) Family with c.4165_4166delAG (p.Gln1388_Ser1389ins*) in *BRCA1* and c.709C>T (p.Arg237*) in *RAD51C.* Index cases are indicated by an arrow. Confirmed mutation carriers are indicated by a “+” sign, and non-carriers are indicated by a “-” sign. Age at diagnosis and cancer type is specified as follows: BC, breast cancer; bBC, bilateral breast cancer; OC, ovarian cancer; CRC, colorectal cancer; PrC, prostate cancer; PaC, pancreas cancer; n.a., not available. The colors stand for each type of cancer as follows: red stands for BC, blue stands for OC, green stands for CRC, yellow stands for Gastric Cancer, black stands for another type of cancer.

**Table 1 ijms-23-11499-t001:** Probands with two pathogenic/likely pathogenic variants.

Family	Proband Phenotype and Age at dx	Gene	Pathogenic Variant DNA Change (Protein Change)	Evidence for P/LP Classification (n° Submissions)			First Degree Cancer Relatives Type and Age at dx	
				ClinVar	LOVD	Our Series	Carriers	Unknown Carriers
678	BC 48y CDI RRHH-	*BRCA1*	c.34C>T (p.Gln13*)	P (18)	P (22)	1	OC 34y HGSC	Esophagus, Gastric
		*BRCA2*	c.1587delTinsCA (p.Glu532Argfs*3)	P (2)	P (3)	5	BC 40y RRHH+/HER2- Gastric 54y; CRC 51y	
776	BC 40y IDC RRHH+/HER2- CRC 54y	*BRCA2*	c.5146_5149delTATG (p.Tyr1716Lysfs*8)	P (10)	P (8)	26	PrC 78y	Sarcoma
		*PMS2*	c.903G>T (p.Lys301Asn)	LP (7)	LP/P (9)	1		2 BC, Bladder
3699	BC 45y IDC RRHH-	*BRCA1*	c.-232_*1383{0} (p.Met1ValfsX13)	-	P (31)	1		OC, CRC, BC, 2 ORL, Lung, glioblastoma
		*BRCA2*	c.5796_5797delTA (p.His1932Glnfs*12)	P (14)	P (20)	1		
C1423	OC 81y HGSC	*BRCA1*	c.4165_4166delAG (p.Gln1388_Ser1389ins*)	P (14)	P (43)	6		PaC, PrC, 3 CRC
		*RAD51C*	c.709C>T (p.Arg237*)	P (14)	P (3)	1		

Phenotypes in double heterozygote families. Abbreviations: dx, diagnosis; y, years-old; P/LP, pathogenic/likely pathogenic; BC, breast cancer; IDC, infiltrating ductal carcinoma; OC, ovarian cancer; HGSC, high-grade serous-papillary ovarian carcinoma; CRC, colorectal cancer; PrC, prostate cancer; PaC, pancreas cancer; ORL, oral cancer; RRHH, hormonal receptors (estrogen and progesterone); HER2, human epidermal growth factor receptor 2; (-) negative; (+) positive. * nonsense mutations; the nucleotide substitution introduces premature termination codon.

## Data Availability

All data generated or analyzed during this study are included in this published article.

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
