# Peer review of "Increased Co-Occurrence of Pathogenic Variants in Hereditary Breast and Ovarian Cancer and Lynch Syndromes: A Consequence of Multigene Panel Genetic Testing?"

_ijms, 2022, doi:10.3390/ijms231911499_

Round 1
Reviewer 1 Report
In this report, the authors reveal the presence of double heterozygosity in familial tumors by analysis using next-generation sequencers. The data are clear and understandable. The text is clear and easy to understand. Also, authors make important clinical recommendations. I think the content is almost acceptable for a case report.
Minor comments:
1. In the Methods section, the authors define the minimum coverage as 30x. Considering the detection of false positives due to PCR errors, coverage should be more than that. If this value is sufficient, please explain this.
2. Please provide a list of the 35 genes sequenced in this study.
Author Response
We thank the Reviewer for the suggestions, and we have modified the revised version of the paper to clarify the Reviewer's comments:
In response to point 1. We have rephrased the sentence to better explain the minimum coverage of 30x. We have changed in line 107 as follows: “Although a minimum coverage of 30x was considered, a mean percent target coverage at 50x was 88.6% and the coverage uniformity was higher than 90% in all tested samples [10].
In response to point 2. We have added a sentence in line 103 of the modified version of the manuscript "(the list of 35 genes is described in Velazquez et al, [10])", in our previous paper were defined the genes included in the customized On-demand Panel.
Reviewer 2 Report
Dear author
Thank you for the submission of your article to our journal. I’ve just read your article and felt some problems αs follows;
- Why do you use question mark in the title?
- You should make clear in your abstract the clinical implications of searching for DH as well as stating that DH will increase in the future.
- You should clarify the significance of publishing these four cases as case reports.
Author Response
Response to question 1. Why do you use question mark in the title?
We appreciate your comment, the interrogative title has been used to trigger a reflection about if the increase of papers reporting double carriers of pathogenic mutations in hereditary cancer genes is the result of the regular use of NGS in clinical practice. Furthermore, highlighting the value of the technique chosen to detect carriers, since as genome-wide studies become more affordable for implementation in clinical practice, the number of double carriers identified will increase even more, allowing personalized treatments for each cancer patient.
Response to question 2. You should make clear in your abstract the clinical implications of searching for DH as well as stating that DH will increase in the future.
The abstract has been modified as suggested, to include clinical implications and how the number of DH detected will increase in the future.
Response to question 3. You should clarify the significance of publishing these four cases as case reports.
The significance of reporting PVs' co-occurrence is, firstly, because it could be an explanation to consider in some patients and, secondly, because disclosing genetic and clinical data of DH patients is key to establish relationships of double mutations in cancer-associated genes with worse clinic or an early onset of cancer.
In that sense we have added two sentences at lines 350 and 355 of the manuscript “Thus, the PVs' co-occurrence should be considered as an explanation for the phenotype displayed in some cases such as our BRCA2/PMS2 patient”, and “…as suggested by others [6] disclosing genetic and clinical data of DH patients is key to establish relationships of double mutations in cancer-associated genes with worse clinic or an early onset of cancer. Notwithstanding, our results should be validated in larger cohorts for more precise results.”
Furthermore, we have added at line 375 in Conclusion section the sentence “… is needed to offer the most appropriate and personalized treatments and surveillance options for each cancer patient in the near future.”
Round 2
Reviewer 2 Report
Dear author
Thank you for the re-submission of your article. I've checked the revised manuscript and confirmed the problems I had pointed out at the initial review properly solved.